



# High frequent SCADA-based thrust load modeling of wind turbines

Nymfa Noppe[1,2], Wout Weijtjens[1,2], and Christof Devriendt[1,2]

[1]Vrije Universiteit Brussel, Pleinlaan 2, 1050 Brussel, Belgium
[2]Offshore wind infrastructure lab (OWI-lab)

*Correspondence to:* Nymfa Noppe (nymfa.noppe@avrg.be)

**Abstract.** A reliable load history is crucial for a fatigue assessment of wind turbines. However, installing strain sensors on every wind turbine is economically not feasible. In this paper, a technique is proposed to reconstruct the thrust load history of a wind turbine based on high frequent SCADA data. Strain measurements recorded during a short period of time are used to train a neural network. The selection of appropriate input parameters is done based on Pearson correlation. Once the training is done, the model can be used to predict the thrust load based on SCADA data only. The technique is validated with both simulation data (FAST) and measurements at an offshore wind turbine. In general, the relative error barely exceeds 15% during normal operation.

## 1 Introduction

As the older wind farms slowly reach their designed lifetime, topics concerning fatigue, remaining useful lifetime and a possible lifetime extension gain importance. Moreover, as fatigue is a design driver for current offshore wind farms, fatigue analysis of existing wind turbines can optimize future design. Currently, fatigue assessments of support structures are often based on measurements of the load history (Loraux and Brühwiler, 2016; Iliopoulos et al., 2017; Schedat et al., 2016; Ziegler et al.). Most of them imply continuous strain measurements at accessible locations. However, the majority of wind turbines will never have strain gauges on them. An alternative approach consists in using load estimations instead of measurements. An important load acting on a wind turbine, both onshore and offshore, is the thrust load, as induced by variations in wind speed. Existing approaches to estimate thrust loads are based on simulations and additional design information (e.g. thrust coefficient) or acceleration measurements (Baudisch, 2012; Cosack, 2010).

As SCADA data is available for every wind turbine by default, several authors (Hofemann et al., 2010; Vera-Tudela and Kühn, 2017) have suggested to use it to estimate the loads on the blades. The estimated model can be translated to every turbine in the farm without the need of installing additional sensors. For this research both 10min and 1s SCADA data is available. Therefore the authors of this contribution propose to use 1s SCADA data to estimate the thrust load, acting on the wind turbine and its substructure.

Although the authors will focus on the estimation of the thrust load solely, additional loads with higher frequencies contribute to fatigue as well. These additional loads are the result of rotor harmonics (3p, 6p, 9p and in case of a rotor imbalance 1p) and structural dynamics (first and second mode, FA1 and FA2). Figure 1a shows the frequency spectrum of measured bending moments, which illustrates the presence of the harmonics as well as the structural modes at an operational wind turbine. In case



of offshore wind turbines, an additional load is induced by waves which is unrelated to any SCADA parameter. Wind turbines installed on monopiles are more affected by waves than those installed on jacket substructures. An approach using SCADA data and accelerometers is proposed by Noppe et al. (2016) to account for the higher frequent loads as well. Although, it was concluded an improvement of the quasi-static model was needed. An alternative approach consists in using a reduced finite
element model of the wind turbine and its substructure (Hartmann and Meinicke, 2017). Here, acting thrust loads are estimated using wind estimator software, which is not publicly available.

As explained, the thrust load has an important contribution to fatigue. However, it is also possible to associate the thrust with properties of wake flows. Therefore, an accurate estimation of thrust also proved important in estimating wake wind speeds and turbulences (Réthoré, 2006).

## 2   Measurement Campaign

### 2.1   Monitoring setup

To validate the proposed technique, results are shown using measurements taken at an offshore wind turbine. The monitored turbine is installed on a jacket and instrumented with strain gauges at the interface between transition piece and tower (Figure 1b). The measured strains are converted into bending moments in fore-aft and side-side direction using the turbine yaw in
the SCADA. The quasi-static contribution of the thrust load to the measured bending moment $M_{tn,m}$ is obtained by using a Butterworth filter of 4th order on the recorded bending moments in a frequency range from 0 to 0.2 Hz. This is shown by the red dashed line in Figure 1a. As the resonance frequency of the first order is 0,31 Hz for this turbine, the contribution from the structural dynamics and the harmonics is excluded from the measurements.

The resulting signal is then transformed into thrust load $F_{t,m}$, using the distance between the sensors (= location of the mea-
sured bending moment) and the hub (= location of acting force) (Réthoré, 2006). To match the time-steps of the SCADA data, the obtained thrust load is down-sampled to time frames of 1 second and 10 minutes.

As the turbine is installed on a jacket, the role of wave loading in the bending moment is assumed to be negligible.

### 2.2   SCADA data

Every wind turbine is installed with a Supervisory Control And Data Acquisition (SCADA) system. The main purpose of
the SCADA system is to monitor and control plants, for which reason it records continuously. The main advantage of using SCADA data is the default availability. However, the correct calibration and quality of the sensors is not guaranteed over the entire lifetime. A common example is the anemometer to measure wind speeds and wind directions. It is installed behind the rotor and known for its high uncertainties due to poor calibrations. Moreover, the quality and accuracy of the data can differ among the different manufacturers. A proper preprocessing of the SCADA data and associated filtering process is advised. In
this case, the preprocessing and filtering process consisted in exclusion of improbable and unrealistic values and periods of constant wind speed from the dataset.





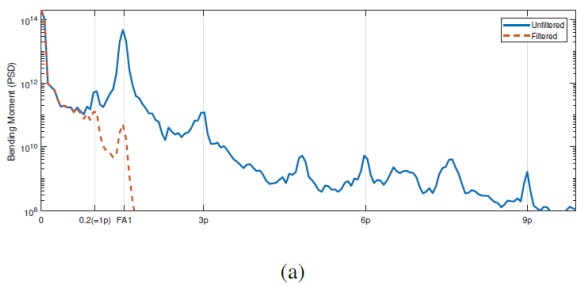
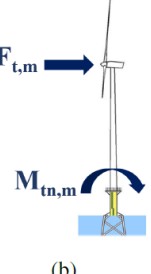

(a)                                          (b)

**Figure 1.** (a) Frequency spectrum of measured tower bending moment in fore-aft direction during 10 minutes (blue line). The quasi-static part of the bending moment is filtered out (red dashed line).

(b) Location of strain gauges (indicated as circles) at the instrumented offshore wind turbine.

For this research a subset of both 10 min and 1s SCADA data was available. The subset consisted in both cases of measurements for wind speed, rotor speed, generated power, blade pitch angle, yaw angle and ambient temperature. Figure 2a shows the power curve obtained with 1s and 10 min SCADA, respectively in blue and purple. The lines indicate the median value of the dataset, while the surface spans from the 5th to the 95th percentile of the data. The power curve shows a much higher variability for 1s

SCADA then for 10min SCADA. The same difference in variability can be observed in Figures 2b and 2c, where 1s and 10min averages of measured thrust load are plotted versus 1s and 10 min SCADA parameters respectively. The present variability in 1s data is not only the result of noise, but is mainly due to the inertias within the controlling system and the wind turbine. For example, when the wind speed increases, the power output increases only a few seconds after. These inertias result in time delays up to several seconds between e.g. the wind speed and the generated power. These delays are not considered constant

over time and will differ for every SCADA parameter. Moreover, they last for only a couple of seconds and in consequence they can not be observed within 10 minute averages.

## 2.3   Meteorological data

Measurements of air pressure are available from a nearby met mast (15km). Using the ambient temperature (from SCADA dataset), the air density $\rho$ is calculated using Equation 1, where p is air pressure (Pa), T is ambient temperature (K) and

$R_{specific}$ is the specific gas constant for dry air (287,058 $\frac{J}{kg \cdot K}$).

$$\rho = \frac{p}{R_{specific}T} \tag{1}$$





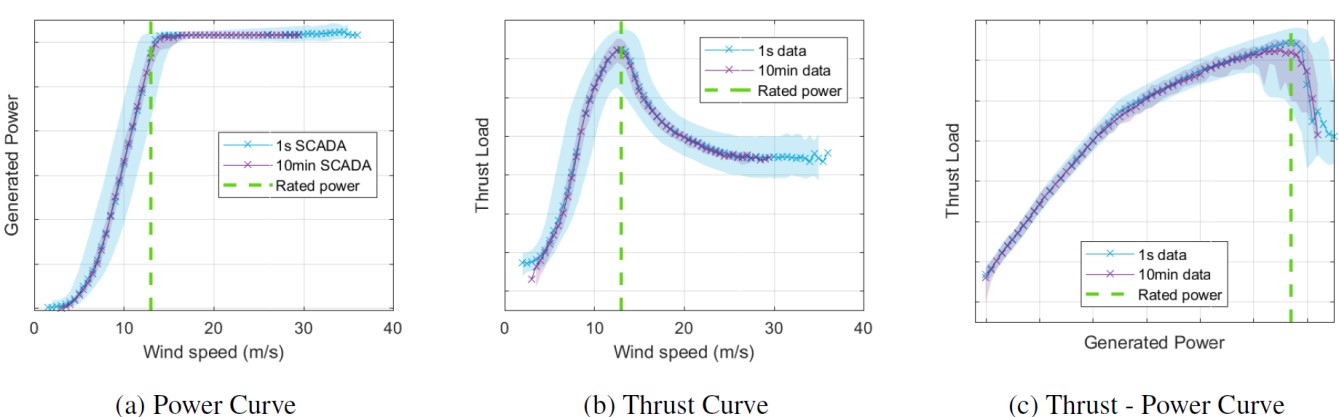

| (a) Power Curve | (b) Thrust Curve | (c) Thrust - Power Curve |

**Figure 2.** Characteristical curves obtained using SCADA data in combination with averages of thrust load measurements. Operational data for a period of 2,5 months is shown. Data in both 10 minute time frame (blue) and 1 second time frame (purple) is shown. The line indicates the median value, calculated per bin of 0,5 m/s or 100 kW, whereas the surface spans from the 5th to the 95th percentile of the data. Rated power is reached for wind speeds of approximately 13 m/s (indicated by the green dashed line).

## 3 Input parameter selection

### 3.1 SCADA data

A crucial part in the model creation is the parameter selection. Input parameters are chosen based on their Pearson correlation to the thrust load. The Pearson correlation between a thrust signal and all considered SCADA signals is calculated using
Equation 2, in which $\bar{X}$ is the mean value of the signal X (May et al., 2011).

$$R(X,Y) = \frac{\sum_{i=1}^{n}(x_i - \bar{X})(y_i - \bar{Y})}{\sqrt{\sum_{i=1}^{n}(x_i - \bar{X})^2 \sum_{i=1}^{n}(y_i - \bar{Y})^2}} \tag{2}$$

This is done for operational data only, both 1s data and 10min data, for a period of 2,5 months. During this period the full wind speed range is covered, as shown by Figure 2. Additionally, the data sets are divided into two subsets: data when the turbine was operating below rated operation (64% of 10min and 62% of 1s operational data) and at rated operation (36% of
10min and 38% of 1s operational data).

The resulting Pearson correlation between the measured thrust load and several SCADA parameters for all data sets is depicted in Figure 3.

Focusing first on the results for the total operational data set, a high correlation can be found for rotor speed (0,8870 and
0,8817 for 10min and 1s data respectively), generated power (0,7516 and 0,7278) and to a lesser extent wind speed (0,5085 and 0,4408). The same conclusion is drawn for operational data below rated power. In this case correlation values are even higher (0,9380 and 0,9595 for rotor speed, 0,9396 and 0,9586 for power, 0,9417 and 0,9028 for wind speed). On the contrary,





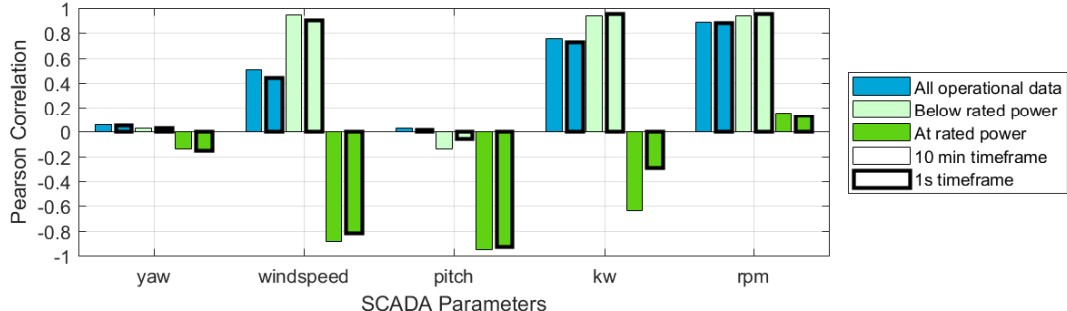

**Figure 3.** The Pearson correlation is calculated between thrust load averages and 5 standard SCADA parameters for operational data only. Both 10 minute and 1s averages are considered. The total dataset is divided based on operational state of the turbine (operating below rated power and operating at rated power).

operational data at rated power reveals a high correlation of the blade pitch angle (0,9501 and 0,9296) and wind speed (0,8901 and 0,8186). This difference in behavior is explained as follows. Once the turbine reached its rated power value, the only parameter acting to varying wind speed and thrust load will be the blade pitch angle. Hence a significant lower correlation for the rotor speed (0,1558 and 0,1389) is found. However generated power is still correlated to thrust load with a significant value

in case of 10 minute averages (0,6363). Figure 2c reveals a very steep curve between thrust and generated power once rated power is reached.

In the results (Figure 3) negative values are the result of an additive inversed relationship between the depicted parameter and the thrust load. For a turbine operating below rated power, a higher wind speed results in a slightly lower blade pitch angle and an increased thrust load. Therefore a decreasing blade pitch angle (due to an increase in wind speed) leads to a

higher thrust load. Hence, a negative value for Pearson correlation between pitch angle and thrust load when the turbine is operating below rated power is expected. Once rated power is reached, increasing wind speeds result in higher blade pitch angles, slightly increasing generated power and decreasing thrust loads (Figure 2b). Accordingly an increase in blade pitch angle and generated power (thanks to an increase in wind speed) enforces a decrease in thrust load. And thus, a resulting negative Pearson correlation between thrust load and wind speed, generated power and pitch angle for operational data at rated

power is consistent.

It is obvious the turbine reacts differently to varying wind speeds depending on the operational state. Once rated power is reached, the relation of the thrust load to the depicted SCADA parameters often differs. This leads to lower correlation values for the total dataset in comparison to the operational states separately.

In general the correlations are less considering 1s averages compared to 10 minutes averages. This can be explained by the

present time delays of several seconds between parameters, as a result of the inertias present within the system.

For the continuation of this research, only the yaw angle will not be considered as an input parameter due to its small correlation with the thrust load.





## 3.2 Meteorological data

According to Baudisch (2012), thrust loads are influenced by air density. While changes in the depicted SCADA variables happen within seconds, air density changes on a different time scale (several hours). Instead of including air density in the set of input parameters, it is accounted for as a correction of the modeled thrust load $\hat{F}_T$: $\hat{F}_{T,corr} = \rho \hat{F}_T$.

## 4 Modeling Method

Seeing the relation between thrust load and the depicted SCADA parameters is non-linear, a model will be created using a neural network. A neural network is capable of finding and characterizing non-linear dependencies within datasets. Therefore it can handle the inverted relations between thrust load and the considered SCADA parameters once rated power is reached. The neural network used in this paper has 3 hidden layers with 4 neurons each. It is trained using operational data only, both while operating below and at rated power. The training data consisted of 1s SCADA data and 1s averages of thrust load measurements ($F_{T,m\_training}$). The input parameters chosen are wind speed, blade pitch angle, rotor speed and generated power, as concluded in Section 3. To account for the inertias in the system, not only instantaneous SCADA values, but also the values of 5 previous seconds are included in the model. Since the model will be used in every normal operational state of the wind turbine, it is important the full operational wind speed band is covered in the training data set. For every operational state, e.g. during a down-rating or curtailment, that is not represented in the training data, the model will probably not be able to predict the thrust load correctly.

To train the neural network, the Neural Network toolbox of MATLAB is used with the default settings (Guide, 2002). This means the preprocessing is done by a min-max mapping function, tan-sigmoid transfer functions are used for hidden layers and a linear transfer function is used for the output layer. Furthermore the data chosen to train the model is randomly divided into 70% of training data, 15% of validation data and 15% of test data. Training is done using the training data and the Levenberg-Marquardt algorithm. Training of the network is stopped when the error on the validation data failed to decrease for 6 iterations or a maximum number of 1000 iterations is reached. The test data is used as an independent dataset of the network training, to calculate the final model error.

As explained in Section 3.2, the effect of air density is accounted for by applying a correction on the model results. To make sure the effect of air density is not present in the training data, the inversed correction is applied on the measured thrust loads of the training dataset: $F_{T,m\_training} = \frac{F_{T,m}}{\rho}$

## 5 FAST Simulations

### 5.1 Simulations

The modeling method proposed in Section 4 will be applied on both simulated data and real-life data. The simulated data is obtained by using the software FAST v8 (Jonkman and Jonkman), offered by NREL. The chosen simulated turbine is the





| Parameter | Category | Description | Unit |
|---|---|---|---|
| Wind1VelX | InflowWind | Nominally downwind component of the hub-height wind velocity | m/s |
| BldPitch1 | ElastoDyn - Blade Pitch Motions | Blade pitch angle (position) | deg |
| LSSGagVxa | ElastoDyn - Shaft Motions | Low-speed shaft strain gage angular speed (on the gearbox side of the low-speed shaft | rpm |
| YawPzn | ElastoDyn - Nacelle Yaw Motions | Nacelle yaw angle (position) | deg |
| TwrBsMyt | ElastoDyn - Tower Base Loads | Tower base pitching (or fore-aft) moment (i.e., the moment caused by fore-aft forces) | kNm |
| GenPwr | ServoDyn - Generator and Torque Control | Electrical generator power | kW |

**Table 1.** The selected output variables for FAST simulations

NREL 5.0 MW Baseline Wind Turbine, installed on an OC3 Monopile RF configuration. All simulation specifications are kept as proposed by the software for use of this turbine type. This means that turbulence and irregular waves are also accounted for. To make sure the full wind speed range is sufficiently covered in the simulation data, several input wind files with varying average wind speed between 3 and 25 m/s are generated using TurbSim. Each wind speed is accounted for equally. In essence, the wind distribution is thus considered as uniform.

The output parameters of interest for this research are specified in Table 1. As the results obtained using simulated data will be used to be compared to real-life data, only comparable parameters for the SCADA data and the measured bending moment are worked with and indicated in Table 1.

The air density is kept constant during the simulations. Therefore the applied corrections for air density didn't influence the results.

## 5.2 Results

To train the model a dataset with a total simulated time of ca 2,5 days is used. Results are shown in Figure 4. A good match between modeled thrust load $\hat{F}_T$ and simulated thrust load $F_{T,s}$ can be found (Figure 4a). Figure 4c shows the relative error $\Delta\epsilon$ between simulated and modeled thrust load ($\Delta\epsilon = \frac{abs(F_{T,s} - \hat{F}_T)}{F_{T,s}}$) versus wind speed for the test set. The line indicates the median value of the relative error, calculated for each wind speed bin of 0,5 m/s. The surface shows 90% of the data, excluding 10% of outliers. In general, the relative error of the independent test set barely exceeds 10%, except for very low wind speeds. A similar behavior was found for the training and the validation set. This indicates the training set was representative for the validation set and the test set.

The model is validated on an additional simulated dataset with a total simulated time of ca. 1,5 days. This data is not used to train the model and can thus be used as a complete independent validation set. Four time series spanning 10 minutes are shown in Figure 5, two while operating below rated power (Figure 5a,c) and two while operating at rated power (Figure 5b,d). The time series of 10 minutes with the highest averaged absolute error between simulated and modeled thrust loads below rated and at rated power are shown in Figure 5c and Figure 5d respectively. In all cases a very good match is found. This is represented

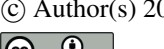



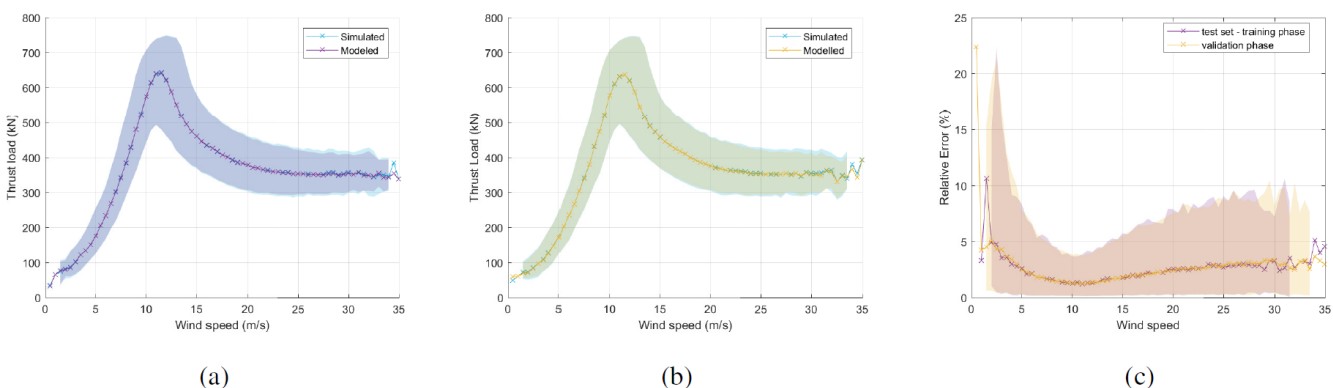

**Figure 4.** Modeled and simulated thrust loads for the training data (training, validation and test set combined, simulated time of 2,5 days)(a) and the long term validation data (1,5 days)(b). The relative error (c) for the test dataset during the training phase and the additional dataset during the validation phase (simulated time of 1,5 days) is shown as well. The lines indicate median values for every wind speed bin of 0,5 m/s; the surface spans from the 5th to the 95th percentile of the data.

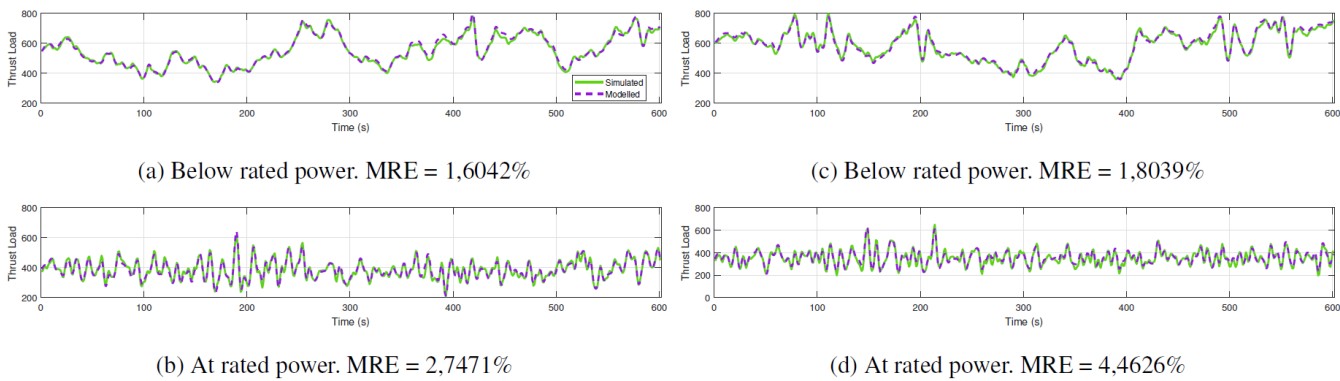

**Figure 5.** Time series (spanning 10 minutes) of modeled and simulated thrust loads below rated (a,c) and at rated power (b,d). For each operational state, the time series with the highest averaged absolute error is shown (c,d). MRE shows the averaged absolute relative error over 10 minutes.

by a low value for the averaged absolute relative error for those time series (MRE < 4,5%). A good match between modeled and simulated thrust loads is found in Figure 4b. Figure 4c shows the relative error versus wind speed, for 90% of the additional validation dataset. Results are similar to those obtained for the test set during the training phase. In general, relative errors do not exceed 10%. For low wind speeds, a higher relative error is found due to the lower absolute values of the thrust load. For higher wind speeds, errors are increasing.





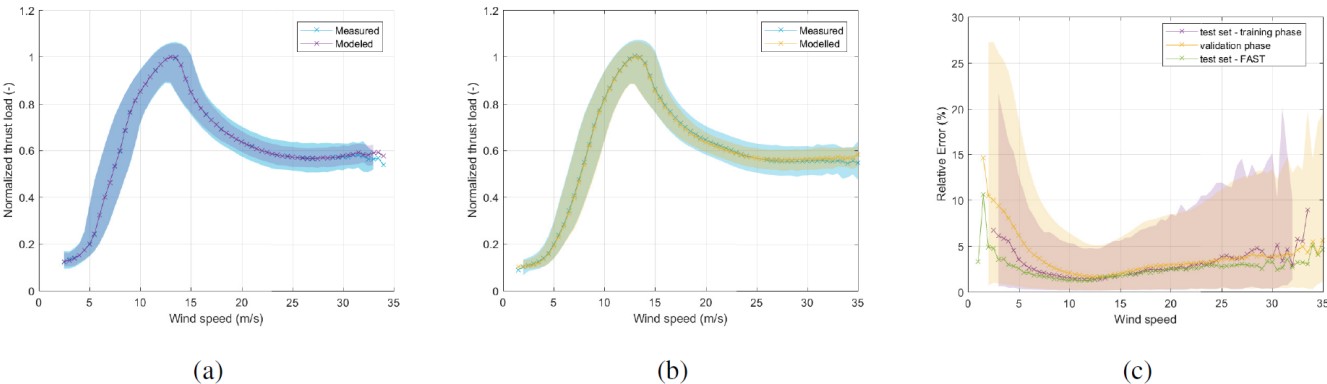

|        (a)        |        (b)        |        (c)        |

**Figure 6.** Modeled and measured thrust loads for the training data (training, validation and test set combined, being 2 weeks worth of data)(a) and the long term validation data (one year worth of data)(b). The relative error (c) for the test dataset during the training phase and the dataset of one year during the validation phase is shown as well. The median values for relative errors obtained with the test set of FAST simulations are copied.

## 6 Application to real world: offshore wind turbine

The proposed modeling method is also tested on an operating wind turbine. Thrust loads are measured using strain gauges, installed at the interface between tower and transition piece, as explained in Section 2. A model is trained using 2 weeks of 1s data. Results are shown in Figure 6. A good match between measured thrust loads $F_{T,m}$ and modeled thrust loads $\hat{F}_T$ can be

found, although above roughly 18 m/s, the modeled thrust curve shows less variability than the measured curve (Figure 6a). Figure 6c shows the relative error ($\Delta\epsilon = \frac{abs(F_{T,m} - \hat{F}_T)}{F_{T,m}}$) of the test set, being 15% of the total training set of 2 weeks. Again the line indicates the median value, calculated for every wind speed bin of 0,5 m/s. The surface spans from the 5th percentile to the 95th percentile of the data. A similar behavior among the training, validation and test set was obtained. In general, the relative error does not exceed 15%. In general, the errors are increased with respect to the results using FAST (as shown in

Figure 6c). The errors obtained for lower wind speeds are increased due to observed offsets in the results. Furthermore, the errors obtained for higher wind speeds are slightly higher due to the loss of variability in the tail of the thrust curve.

This model is validated on a dataset of one year, including the two weeks of training data. Four time series of 10 minutes are shown in Figure 7. Two of them show operation below rated power (a and c), while the other two show operation at rated power (b and d). The time series depicted in Figure 7c and Figure 7d show 10 minutes with the highest averaged absolute error

when operating below or at rated power respectively. An offset can be observed in (a,c), while the loss of variability can be observed in (d). The values of the averaged absolute relative error indicate the match is still acceptable (MRE < 6,5%). A good match between modeled and measured thrust loads is found in Figure 6b. Again loss in variability can be observed for wind speeds higher than approximately 18 m/s. Figure 6c also shows the relative error obtained during one year of operation versus the wind speed, for 90% of the data points. With an median value barely exceeding 5%, results are promising. For higher wind

speeds, results are similar to those obtained during the training phase. For wind speeds up to 10 m/s, on the other hand, errors

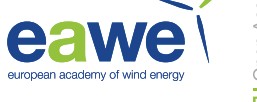
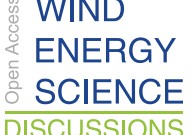

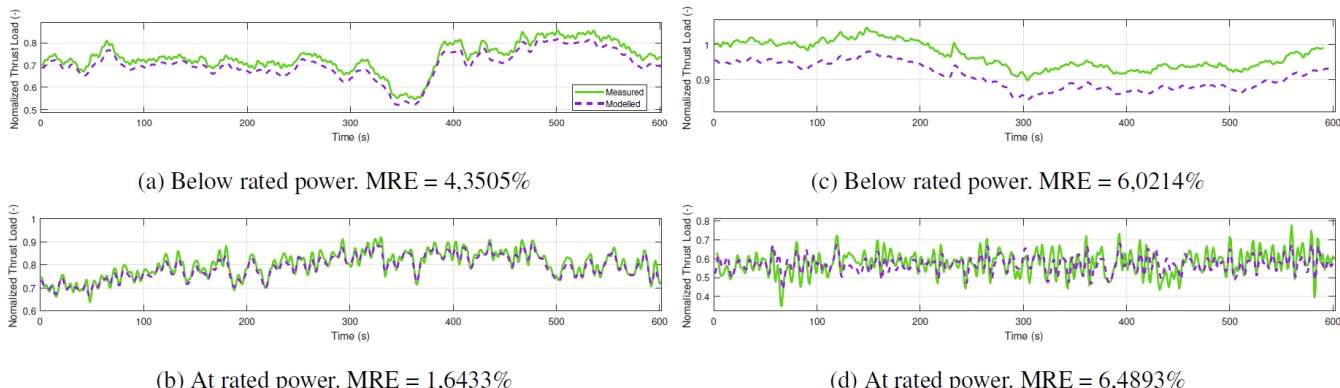

(a) Below rated power. MRE = 4,3505%

(c) Below rated power. MRE = 6,0214%

(b) At rated power. MRE = 1,6433%

(d) At rated power. MRE = 6,4893%

**Figure 7.** Time series (spanning 10 minutes) of modeled and measured thrust loads below rated (a,c) and at rated power (b,d). For each operational state, the time series with the highest averaged absolute error is shown (c,d). MRE shows the averaged absolute relative error over 10 minutes.

in the validation phase are higher than those of the test set during the training phase, due to bigger offsets. However, as fatigue is driven by cyclic loading, these offsets won't influence a fatigue assessment.

Calculating the Pearson correlation between measured and modeled thrust load for the same period of 2,5 months (cfr. Section 3) results in 0,9957. This high value indicates a lot of the present variability in the thrust load can be explained by combining several SCADA parameters. When correlating thrust load to only one SCADA parameter, the present scatter will partially hide the correlation. Whereas a non linear combination of multiple parameters is used, as is done by the neural network, a higher correlation can be found.

## 7 Conclusions

An approach to estimate thrust load signals based on SCADA signals is explained and validated both on simulation and real measurement data. Wind speed, rotor speed, blade pitch angle and generated power are selected as input parameters based on a linear correlation analysis. Strain sensors are used to measure the acting thrust load. This thrust load signal is combined with SCADA signals to train a neural network. Validation of the method is done using FAST simulation data and data measured at an offshore wind turbine during one year. Time series show a good match between modeled and measured or simulated thrust signals. In general, the relative error barely exceeds 15%. Results obtained using FAST data are slightly better than those of the real world offshore wind turbine.

Essential in this approach is the preprocessing of the SCADA data. Moreover, including an air density correction proved to reduce offsets in the results. Furthermore good results are obtained even when using the default settings of the neural network. Adjusting the neural network hyperparameters, e.g. number of layers and neurons, did not improve the results significantly.





## 8 Future Work

The use of SCADA data can be considered as the main advantage of this approach. If the model proofs to be transferable among turbines of the same type, this approach can be applied on any (non-instrumented) wind turbine within a wind farm. This transferability should be validated by cross validation among instrumented turbines.

The presented method is capable of estimating quasi-static loads on wind turbines. To perform a full fatigue assessment of wind turbines, structural and rotor dynamics have to be accounted for as well. For offshore wind turbines with significant wave-loading, e.g. large diameter monopiles, also the effect of waves on the structure needs to be included. A full load reconstruction can be done by combining the proposed approach with acceleration measurements (Noppe et al., 2016).

*Acknowledgements.* This work has been funded by the Institute for the Promotion of Innovation by Science and Technology in Flanders (IWT) in the framework of the VIS OWOME Project and by the Research Foundation - Flanders.



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
