# Peer review of "High frequent SCADA-based thrust load modeling of wind turbines"

_Wind Energy Science, 2017_

## Referee Comment (RC1) · Anonymous Referee #1 · 8 Nov 2017

A well written manuscript that details the estimation of thrust loading on wind turbines using 1s SCADA data.

Although many, the requests for corrections and clarifications are minor. Positive feedback is also added to point out relevant pieces in the contribution. Congratulations to the authors on such good work!

Thus, I add a notation to facilitate working the comments: *must, +clarify, -minor, !great.

All the best,
* * *
- title, it states 'high frequent' whereas it uses 1s; somehow this could be made explicit,

you never know if another person is already working on 10Hz, so make it part of the title '... frequent (1s) ...'

+ abstract, expand if possible by the number of words, otherwise in the introduction, to justify why were the simulations needed and under which criteria 15% error is assessed (both in time and quality). I kind of infer from the lack of direct measurement, but it is better if explicit.

- introduction, explain the, typically non-written, motivation behind the use of 1s, which a priori represent more information over 10m data; was it defined according to methodology, simply accessible or the like.

+ introduction, modelling in a time-series domain instead over 10m statistics have advantages. What goals did you envisioned? Where is the knowledge gap?

+ page 1, line 21, is 10m SCADA meant sampled data or statistics?

+ page 2, line 30, more information about the filtering process is needed. Although it is data dependent and designer driven, a fair reproduction from the manuscript will be impossible without this step. Please indicate what %-data were removed.

- page 3, Fig. 1, increase size of fonts (a), circles in (b) did not show in print.

+ page 3, line 1, indicate how many samples did the data contain or its duration if record was continuous.

+ page 3, line 9, (also page 5, line 19) either here or in the discussion, time-lag analysis would have indicated the statistical duration of lag between series. It is of interest to understand its impact on the model (see page 6, line 13 regarding the use of previous records) and on the estimation of thrust (excitation) after moments at the bottom of the tower (response) is recorded.

!+ page 4, Fig 2c, looking at it reminds me of heteroscedacity (even more when looking at Fig 4c and 6c). Have you consider it? I think is of great value that you included

residuals next to results! It helps to understand limitations of the model, bring it to the discussion.

- page 4, line 4, Pearson was used, OK fair, but why not to consider Spearman or information coefficient to account for non-linearities?

- page 6, line 9 (then line 13), was the architecture and the number of previous records optimized? are they used in all channels?

- page 6, line 11, when the inputs are mentioned it is now clear that 10-minute data was only used to explore correlations, or I did get it wrong?

- page 6, line 20, without knowing the number of data points in your data set, the question is why was hold-out preferred over cross-validation? check Hastie, the elements of statistical learning for a discussion.

* page 7, line 2, a reference next to '... proposed by the software...' is needed.

* page 7, line 2, if I understood correctly, simulations were meant to be site-specific, please share more information about the model used and whether it was validated or not.

- page 7, line, the comparison of measurements to simulations is not clear in the overall aim of the manuscript (a statistical description to assess if both processes behind the data are similar or not will lead to other focus), it does not make it worst, but when the reader misses the point, it might have missed the relevance of such an effort

+ page 7, line 14, one of the weakness (to increase the rate of the paper) is that estimations are verified versus simulated values, which rises the question on its relevance. Would the evaluation assess measured $M_{tn,m}$ data as a previous step, then the inductive procedure would appear more robust.

- page 16, line 15, the decision to represent 90% of the data, was it for clarity of the graph or for other reason? The use of different language (see Fig 2, referring to

quantiles, rises my question)

- page 9, line 19, there is a typo 'With a(n) median...' just found it by chance.

* page 10, line 2, I don't agree on the statement 'these offsets won't influence a fatigue assessment', as the distribution of offsets and variance is not included in the analysis, this needs to be provided next to its verification. Furthermore, since damage would have an exponent in the operation, a small difference would be increased dramatically.

- page 10, line 19, please give an numerical range to your definition of significance (a reference would be even better), people will understand it differently.

- page 11, line 1, I would rather specify the use of 1s SCADA data.

! the tone of the paper is entertaining and well balanced, congratulations!
* * *

---

## Referee Comment (RC2) · Anonymous Referee #2 · 20 Nov 2017

General comments: The authors are investigating if the quasi-static rotor thrust can be estimated from SCADA signals of a wind turbine. They present interesting studies based on simulated and measured data which are worth to be published. The paper itself is written in appropriate style and form. Nevertheless, there are several issues and questions which should be addressed in more detail before publication.

As a general comment, the authors state correctly that the assessment of fatigue load history is important for the estimation of the remaining useful lifetime of existing wind farms and also to gather knowledge for the optimization of future wind farms. They also mention that fatigue is driven by cyclic loading mostly (page 10, line 2). At the same time the authors are presenting a method for the estimation of a quasi-static thrust load, where most of the load cycles have been removed. This inevitably leaves the reader

with the question why the authors have chosen to estimate the quasi-static thrust load if it is not that relevant for fatigue. The authors should clarify this contradiction and explain in more detail their motivation for the estimation of the quasi-static thrust load.

Moore detailed comments are as follows:

Page 1, Title: The authors may want to revise the title of their paper. I am not a native English speaker, but shouldn't it read 'High frequency...' instead of 'High frequent...'? The authors should further discuss, if 'high frequency'/'high frequent' is not misleading for this paper, as they are modelling the quasi-static component of the thrust only and have removed all 'high frequency' content from the signal. Maybe 'Modelling of quasi-static thrust of wind turbines based on SCADA data' is more accurate?

Page 1, line 23ff: It sounds as if the thrust load would not show oscillations with the multiple of the rotor frequency (1P, 3P, 6P...) or with natural frequencies of the support structure. However, Figure 1a shows that these frequencies are clearly visible in the spectra of the thrust load. The authors should explain this in more detail.

Page 2, line1: Typically there is a correlation between wind speed and wave height or wave period for example. Hence it is not true that wave induced loading is unrelated to any SCADA signal, e.g. the measured wind speed. Please clarify.

Page 2, line 16: In Figure 1a it is shown that the 1P rotor frequency is at 0.2Hz. It is unclear if this is the 1P at rated speed or minimum speed, for example. If it is assumed that this is the 1P at rated speed, then the 1P would be lower for partial load operation, e.g. 0.1 Hz at minimum speed. In this case, the applied filter would not remove the 1P frequency content from the thrust signal. The authors should elaborate on this and explain their approach for selecting the filter frequency more clearly.

Page 2, line17: It is more common to write 'first natural frequency' instead of 'resonance frequency of the first order'. Please consider changing the wording.

Page 2, line 21: It is unclear how the downsampling has been performed. Are signals

averaged over 1 second and 10 minutes or are data points simply removed from the signal to achieve the desired resolution?

Page 4, line 4ff: The Pearson correlation coefficient is a measure for the linear correlation of two signals. If the relation between two signals is non-linear the correlation coefficient may be small. In that respect, a small correlation coefficient could simply indicate that there is no linear relation between the independent and the dependent variable, but it does not mean that there is no relation at all. Hence, even if the Pearson correlation coefficient between the thrust and a SCADA signal is small, the signal still may be valuable input to the neural network. Cuold the authors please explain in more detail why they have chosen to use the Pearson correlation coefficient for the identification of suitable input signals for their estimation of thrust loading with neural networks? Furthermore, on page 3, line 9-11 the authors mention correctly that there may be time delays between the thrust load and the SCADA signals. Can these time delays also result in reduced correlation coefficients for the 1s operational data and if so, how have the authors dealt with this in their investigation?

Page 6, line 6: Similar comment as above: The authors state that the relation between thrust load and SCADA signals is non-linear. At the same time they use a linear correlation coefficient to test is a relation exists. The authors should clarify this contradiction.

Page 6, line 9: Why have the authors chosen this network topology?

Page 7, line 11-18: Figure 4b is not explained in the text. The authors may consider adding a sentence here.

Page 8, Caption of Figure 4: It is unclear why the validation set is denoted as "long-term". Has that data been recorded at a different period of time, e.g. some month later than the training data?

Page 8, line 1-5: It is not clear why Figure 4b is explained here. In addition, Figure 4c has been explained on page 7 already. It may not be necessary to repeat the

explanation here again.

Page 9, Chapter 6: Is it possible to show the probability distribution of wind speeds in the measured data? And have the authors investigated, if the relative error is somehow related to the probability of wind speed? The distribution of the relative error looks similar to a flipped Weibull distribution. Possibly the network was able to learn the relation between thrust load and SCADA data for those wind conditions that were over-represented in the raining data and the relation at very low and very high wind speeds was not learned that well. Have the authors tried to use training data featuring different wind speed distributions and compared the graphs of the relative error?

Page 9, line 10-11: The explanation for the errors at low and high wind speeds is unclear. What are the "offsets in the results" and what is meant by the "variability in the tail of the thrust curve"? Could the authors please explain this in more detail?

Page 9, line 16ff: It is unclear why the authors describe the content of Figure 6 again and also after the content of Figure 7 was explained. The content of Figure 6 was discussed at the beginning of the page before and any additional information in this section may also be moved to the beginning of the page. Page 10, line 6-8: What is meant by "the present scatter will partially hide the correlation"? Do the authors want to state that the correlation coefficient between measured and modelled thrust load is smaller if only one independent variable is used compared to multiple independent variables?

Page 10, line 18: It is unclear what the authors mean by "default settings of the neural network". Have they chosen a default network topology for this study? Please explain in more detail.

---

## Author Comment (AC1) · 22 Dec 2017

Thank you for your nice and interesting comments. We agreed to most of them and already discussed them during the project. Following you can find our answers and how we suggest to adjust the manuscript.

Both reviewers suggested to revise the title of the manuscript. We agree with them and therefore the title will be modified to "Modelling of quasi-static thrust load of wind turbines based on 1 second SCADA data".

From several comments of both reviewers, we realized the motivation behind the thrust

load estimation and the use of 1s SCADA data is not obvious. As 1s SCADA data was accessible for this research and is becoming the standard in industry, the main goal was to explore its possibilities to estimate fatigue loads. Given the sampling frequency of 1 Hz, only low frequent loads (lower than 0,5 Hz) can be estimated. In addition it was soon clear the first order tower dynamics were not present in the SCADA signal. Therefore the load spectrum that can be estimated based on 1s SCADA data is limited to the quasi-static thrust load.
Previous research within our group showed fatigue loading can be estimated using a combination of strain gauges and accelerometers. For several reasons accelerometers are preferred over strain gauges, but they are not suited for quasi-static loads. The strain gauges are thus crucial to capture the quasi-static part of the loading. The research presented in this manuscript aims to replace the strains gauges by the SCADA-based approach. In future research the proposed thrust load estimation can be combined with the use of accelerometers to estimate the total fatigue loading. This motivation will be explained better in the introduction.

Both reviewers asked for clarification regarding the time delays mentioned in the paper. During this research, an autocorrelation was performed between a thrust signal and several SCADA signals for multiple periods. Results showed the observed time shift changed for different signals but also for different periods. The first attached figure (CorrelationCoefficient_vs_timeshift) shows the correlation coefficient between a thrust signal and 4 SCADA signals of 2,5 months, where a time lag of -15 to 15 seconds was considered between the signals. In general, the differences in correlation coefficient are fairly low. We decided the added value of including this figure to the manuscript was small, but based on the comments following sentence will be added to page 5, line 20: *When calculating the autocorrelation between the thrust load signal and shifted SCADA signals, the biggest time shift was found for the pitch signal and corresponded to -3 seconds.*
Since the maxima never exceed 5 seconds, it was chosen to include the 5 previous

timestamps for every selected SCADA parameter as an input to the neural network. Moreover a small sensitivity analysis (trial and error) revealed that including less previous timestamps in the input set of the neural network made the obtained results worse, there was also no gain beyond 5 seconds.

Both reviewers pointed out that Pearson correlation only applies for linear correlations. We fully agree and calculated mutual information too during this research to consider non-linear correlations. Results were comparable to those obtained with Pearson correlation. Since results were comparable and for sake of simplicity it was decided at the time to only include Pearson correlation in the manuscript. Since this rightfully raises questions, the results for mutual information will be included as well in the revised version of the manuscript. The most important observation in these results is that all selected parameters are clearly correlated to the thrust load based on the complete dataset, whereas the Pearson correlation of the pitch angle to the thrust load was low when all data was considered.

Some comments given by both reviewers concerned the topology of the neural network. This topology was chosen by the authors in the beginning of the project. Three hidden layers were chosen because three operational states can be distinguished for a wind turbine ('non-operating', 'operating below rated power' and 'operating at rated power'). Moreover 4 neurons were chosen in each layer because 4 input parameters were selected. The second attached figure (MeanRelativeError_vs_different_topologies) shows the possible gain in mean relative error of the test set by using a different topology. In this case only topologies are considered with the same number of neurons in each layer. These results show the error is not influenced a lot by the topology, as long as more than one neuron is used. The topology was not optimized afterwards because results were already satisfying. Following sentence will be added to the contribution: *By choosing a different topology the mean relative error*

*of the test set improved with maximum 0,2% if more than one neuron is chosen in each layer.*
All the other needed settings for the neural network are kept as suggested by the Neural Network Toolbox of MATLAB.

Based on several comments, we understand that the motivation behind the use of simulations is not fully clarified in the manuscript. During the project we decided to use simulated data, which provided a controlled and reproduceable environment, in order to understand the real measurements better. However the simulated turbine (and site) is not the same turbine, although comparable, as the real one. Therefore the simulated thrust load cannot be compared to the measured thrust load. We included the results based on simulated data to show the method works for different types of turbines and to provide a reproduceable environment to optimize the technique in the future. If desired we can make the simulation data publicly available. Several adjustments will be made to the manuscript. The abstract will be modified to motivate the two-step approach of both simulation and real-world measurements. Moreover section 5 and section 6 will be merged to one section "Results". This section will start with following text:
*The proposed approach is validated using two different datasets. The first one is obtained by simulation in FAST, while the second one is obtained thanks to a measurement campaign performed at an offshore wind turbine. The dataset obtained by simulations was included to illustrate the approach in a controlled and reproduceable environment.*

Smaller comments are addressed in the following.

*"+ abstract, expand if possible by the number of words, to justify under which criteria 15% error is assessed (both in time and quality)."*

The error is assessed between the direct simulations/measurements of the thrust load and the predicted result. The abstract will be adjusted.

"+ page 1, line 21, is 10m SCADA meant sampled data or statistics¿'
Statistics, will be clarified in paper.

"+ page 2, line 30, more information about the filtering process is needed. Although it is data dependent and designer driven, a fair reproduction from the manuscript will be impossible without this step. Please indicate what %-data were removed."
Values outside the interval [mu-3*sigma;mu+3*sigma] are removed. In total, less than 6% is removed. This will be added to the paper.

"- page 3, Fig. 1, increase size of fonts (a), circles in (b) did not show in print."
Font size and caption will be adjusted.

"+ page 3, line 1, indicate how many samples did the data contain or its duration if record was continuous."
The length of the investigated data-set (one year) will be mentioned in the paper.

"!+ page 4, Fig 2c, looking at it reminds me of heteroscedacity (even more when looking at Fig 4c and 6c). Have you consider it?"
This is an interesting point. We'll add a comment on the increased variability of the modelling error with the wind speed to the text. It is however not a path we have investigated further, but we are open for suggestions.

"- page 6, line 11, when the inputs are mentioned it is now clear that 10-minute data

was only used to explore correlations, or I did get it wrong?"
The 10-minute statistics are indeed only used to explore correlations.

"- page 6, line 20, without knowing the number of data points in your data set, the question is why was hold-out preferred over cross-validation? check Hastie, the elements of statistical learning for a discussion."
The training set used for the offshore wind turbine consisted in over one million data points after filtering. Given the big dataset hold-out was preferred to reduce computational cost. Moreover, during the project we decided to use the default settings proposed by the Neural Network toolbox of MATLAB. Since we were already satisfied with the result using these default settings, we chose not to optimize the settings. Therefore, optimization of the method is still possible and using cross-validation instead of hold-out can lead to better results but was not investigated during this research.

"* page 7, line 2, a reference next to '... proposed by the software...' is needed. "
The reference will be added.

"- page 16, line 15, the decision to represent 90% of the data, was it for clarity of the graph or for other reason? The use of different language (see Fig 2, referring to quantiles, rises my question)"
It is exactly the same, the sentence will be rephrased.

"- page 9, line 19, there is a typo 'With a(n) median...' just found it by chance."
Thanks, it will be adjusted.

"* page 10, line 2, I don't agree on the statement 'these offsets won't influence a

*fatigue assessment', as the distribution of offsets and variance is not included in the analysis, this needs to be provided next to its verification. Furthermore, since damage would have an exponent in the operation, a small difference would be increased dramatically."*

We consider a fatigue assessment independent from offsets assuming this fatigue assessment is performed according to common practise in industry. This means performing cycle counting and the Miner's rule. Since this is based on the size of the cycle and independent from the mean level, we concluded offsets won't influence a fatigue assessment. This will also be clarified in the manuscript.

*"- page 11, line 1, I would rather specify the use of 1s SCADA data."*
This will be specified in the paper.

Thank you very much for your nice comments and helpful review!

[Figure]

**Fig. 1.** CorrelationCoefficient_vs_timeshift

[Figure]

**Fig. 2.** MeanRelativeError_vs_different_topologies

---

## Author Comment (AC2) · 22 Dec 2017

Thank you for your nice and interesting comments. We agreed to most of them and already discussed them during the project. Following you can find our answers and how we suggest to adjust the manuscript.

Both reviewers suggested to revise the title of the manuscript. We agree with them and therefore the title will be modified to "Modelling of quasi-static thrust load of wind turbines based on 1 second SCADA data".

From several comments of both reviewers, we realized the motivation behind the thrust

load estimation and the use of 1s SCADA data is not obvious. As 1s SCADA data was accessible for this research and is becoming the standard in industry, the main goal was to explore its possibilities to estimate fatigue loads. Given the sampling frequency of 1 Hz, only low frequent loads (lower than 0,5 Hz) can be estimated. In addition it was soon clear the first order tower dynamics were not present in the SCADA signal. Therefore the load spectrum that can be estimated based on 1s SCADA data is limited to the quasi-static thrust load.

Previous research within our group showed fatigue loading can be estimated using a combination of strain gauges and accelerometers. For several reasons accelerometers are preferred over strain gauges, but they are not suited for quasi-static loads. The strain gauges are thus crucial to capture the quasi-static part of the loading. The research presented in this manuscript aims to replace the strains gauges by the SCADA-based approach. In future research the proposed thrust load estimation can be combined with the use of accelerometers to estimate the total fatigue loading. This motivation will be explained better in the introduction.

Both reviewers asked for clarification regarding the time delays mentioned in the paper. During this research, an autocorrelation was performed between a thrust signal and several SCADA signals for multiple periods. Results showed the observed time shift changed for different signals but also for different periods. The first attached figure (CorrelationCoefficient\_vs\_timeshift) shows the correlation coefficient between a thrust signal and 4 SCADA signals of 2,5 months, where a time lag of -15 to 15 seconds was considered between the signals. In general, the differences in correlation coefficient are fairly low. We decided the added value of including this figure to the manuscript was small, but based on the comments following sentence will be added to page 5, line 20: *When calculating the autocorrelation between the thrust load signal and shifted SCADA signals, the biggest time shift was found for the pitch signal and corresponded to -3 seconds.*

Since the maxima never exceed 5 seconds, it was chosen to include the 5 previous

**WESD**
timestamps for every selected SCADA parameter as an input to the neural network. Moreover a small sensitivity analysis (trial and error) revealed that including less previous timestamps in the input set of the neural network made the obtained results worse, there was also no gain beyond 5 seconds.

Both reviewers pointed out that Pearson correlation only applies for linear correlations. We fully agree and calculated mutual information too during this research to consider non-linear correlations. Results were comparable to those obtained with Pearson correlation. Since results were comparable and for sake of simplicity it was decided at the time to only include Pearson correlation in the manuscript. Since this rightfully raises questions, the results for mutual information will be included as well in the revised version of the manuscript. The most important observation in these results is that all selected parameters are clearly correlated to the thrust load based on the complete dataset, whereas the Pearson correlation of the pitch angle to the thrust load was low when all data was considered.

Some comments given by both reviewers concerned the topology of the neural network. This topology was chosen by the authors in the beginning of the project. Three hidden layers were chosen because three operational states can be distinguished for a wind turbine ('non-operating', 'operating below rated power' and 'operating at rated power'). Moreover 4 neurons were chosen in each layer because 4 input parameters were selected. The second attached figure (MeanRelativeError\_vs\_different\_topologies) shows the possible gain in mean relative error of the test set by using a different topology. In this case only topologies are considered with the same number of neurons in each layer. These results show the error is not influenced a lot by the topology, as long as more than one neuron is used. The topology was not optimized afterwards because results were already satisfying. Following sentence will be added to the contribution: *By choosing a different topology the mean relative error*
of the test set improved with maximum 0,2% if more than one neuron is chosen in each layer.

All the other needed settings for the neural network are kept as suggested by the Neural Network Toolbox of MATLAB.

Based on several comments, we understand that the motivation behind the use of simulations is not fully clarified in the manuscript. During the project we decided to use simulated data, which provided a controlled and reproduceable environment, in order to understand the real measurements better. However the simulated turbine (and site) is not the same turbine, although comparable, as the real one. Therefore the simulated thrust load cannot be compared to the measured thrust load. We included the results based on simulated data to show the method works for different types of turbines and to provide a reproduceable environment to optimize the technique in the future. If desired we can make the simulation data publicly available. Several adjustments will be made to the manuscript. The abstract will be modified to motivate the two-step approach of both simulation and real-world measurements. Moreover section 5 and section 6 will be merged to one section "Results".

The proposed approach is validated using two different datasets. The first one is obtained by simulation in FAST, while the second one is obtained thanks to a measurement campaign performed at an offshore wind turbine. The dataset obtained by simulations was included to illustrate the approach in a controlled and reproduceable environment.

Several comments concerned the order of the figures addressed in the text of Sections 5 and 6. We agree this could be confusing. The order of the figures was chosen to be able to easily compare training and validation phase. However the order of explanation in the text was chosen to make the distinction between the training and validation
phase. Since this confuses the reader, the order of explanation will be adjusted. First the procedure will be explained, being a training phase and a validation phase. Afterwards the resulting relative errors during both phases will be discussed together. Moreover, in case of the simulated data, the wording "long term validation data" was chosen to correspond to the measurements as the complete set was used to validate. However the wording will be adjusted. To end, some timeseries will be discussed to illustrate the obtained relative errors.

Smaller comments are addressed in the following.

Page 1, line 23ff: "It sounds as if the thrust load would not show oscillations with the multiple of the rotor frequency (1P, 3P, 6P. . .) or with natural frequencies of the support structure. However, Figure 1a shows that these frequencies are clearly visible in the spectra of the thrust load. The authors should explain this in more detail."

Figure 1a shows the spectrum of the measured bending moment, this indeed contains the contribution of the rotor harmonics. However the primary target of this paper are the quasi-static loads up 0.2Hz (Filtered). For these specific loads the rotor harmonics do not contribute. To avoid confusion, the legend of Figure 1a will be adjusted. Moreover the sentence "the targeted quasi static load (filtered) does no longer contain the effects of rotor harmonics" will be added to the text and the caption.

Page 2, line1: "Typically there is a correlation between wind speed and wave height or wave period for example. Hence it is not true that wave induced loading is unrelated to any SCADA signal, e.g. the measured wind speed. Please clarify."

We agree a correlation between wind speed and wave height or wave period exists. However this correlation only holds for e.g. 10 minute statistics. However, in timeframes of a couple of seconds, wind speed and wave height or period are not be correlated one on one, e.g. a gust does not imply a higher inbound wave. The Interactive comment

sentence will be rephrased in the paper to clarify this subtilty.

Page 2, line 16: "In Figure 1a it is shown that the 1P rotor frequency is at 0.2Hz. It is unclear if this is the 1P at rated speed or minimum speed, for example. If it is assumed that this is the 1P at rated speed, then the 1P would be lower for partial load operation, e.g. 0.1 Hz at minimum speed. In this case, the applied filter would not remove the 1P frequency content from the thrust signal. The authors should elaborate on this and explain their approach for selecting the filter frequency more clearly."

In general, turbines are well balanced and a 1p rotor harmonic does not exist. The filter frequency was chosen in such a way the filtered signal was not influenced by the first resonance frequency, since this is unrelated to any SCADA signal anyway. However this filter frequency can be adjusted once the approach is combined with accelerometers. In that case the lower frequency bound of the accelerometers (0,06Hz) can be set as the filter frequency for the SCADA-based thrust load estimation. We considered this path outside the scope of this contribution. The motivation behind the filter frequency will be clarified in the paper.

Page 2, line17: *"It is more common to write 'first natural frequency' instead of 'resonance frequency of the first order'. Please consider changing the wording."* True, the wording will be adjusted.

Page 2, line 21: "It is unclear how the down sampling has been performed. Are signals averaged over 1 second and 10 minutes or are data points simply removed from the signal to achieve the desired resolution?"

The sentence will be adjusted to: the obtained thrust load is down-sampled using an antialiasing filter to a time frame of 1 second and additionally averaged over 10 minutes.

**WESD**
Page 9, Chapter 6: "Is it possible to show the probability distribution of wind speeds in the measured data? And have the authors investigated, if the relative error is somehow related to the probability of wind speed? The distribution of the relative error looks similar to a flipped Weibull distribution. Possibly the network was able to learn the relation between thrust load and SCADA data for those wind conditions that were overrepresented in the raining data and the relation at very low and very high wind speeds was not learned that well. Have the authors tried to use training data featuring different wind speed distributions and compared the graphs of the relative error?" This is an interesting comment. We decided not to change the wind speed distribution of the training set since we considered the common wind speeds more important. However, during the project we trained a neural network by using high windspeed data only. This yielded similar results as shown in the contribution. Moreover in case of the simulated data, the wind speed was uniformly distributed and we observed the same distribution of relative error.

Page 9, line 10-11: "The explanation for the errors at low and high wind speeds is unclear. What are the "offsets in the results" and what is meant by the "variability in the tail of the thrust curve"? Could the authors please explain this in more detail?" We called the mean difference between measured or simulated and modelled values the offsets in the results. Moreover for windspeeds higher than ca 15 m/s, the difference between the 5th and 95th percentile of modelled thrust is lower than the difference between the 5th and 95th percentile of measured thrust. The wording will be adjusted in the contribution.

Page 10, line 6-8: "What is meant by "the present scatter will partially hide the correlation"? Do the authors want to state that the correlation coefficient between measured and modelled thrust load is smaller if only one independent variable is used compared to multiple independent variables?"

**WESD**
Yes, however the variables we used are not independent. In this case the thrust load depends on multiple parameters. These parameters show dependencies among them as well. When looking at only one parameter, the difference in value for thrust measurements occurring for the same value of that parameter cannot be explained. When considering more parameters, this difference might already be explained by a different value of another parameter. Therefore the resulting correlation between the measured thrust load and only one parameter is lower than between the measured thrust load and a combination of multiple parameters, as done by the neural network.

Thank you very much for your nice comments and helpful review!
windspeed - operating below rated pitch - operating at rated (inversed)

power - operating below rated rpm - operating below rated

Fig. 1. CorrelationCoefficient\_vs\_timeshift

1

0.98

---

## Editor Comment (EC1) · G. J. W. van Bussel (Editor) · 5 Jan 2018

Dear Authors,

Thank you for seriously addressing the comments and for the way in which you plan to implement the comments in a revised manuscript. Please upload a new manuscript in which you have marked all modifications (see authors instructions) so we can proceed in getting the final version published.
* * *

---

## Author Comment (AC3) · 15 Jan 2018

Dear Gerard and reviewers,

In attachment you can find the modified manuscript, where all changes are marked in blue.

Please also note the supplement to this comment:
https://www.wind-energ-sci-discuss.net/wes-2017-46/wes-2017-46-AC3-supplement.pdf